# Delay in Seeking Medical Attention and Diagnosis in Chinese Melanoma Patients: A Cross-Sectional Study

**DOI:** 10.3390/ijerph192214916

**Published:** 2022-11-12

**Authors:** Xinchen Ke, Tianhao Wu, Guiyun Gao, Songchun Yang, Wenrui Lin, Yi Xiao, Minxue Shen, Mingliang Chen, Xiang Chen, Shuang Zhao, Juan Su

**Affiliations:** 1Department of Dermatology, Xiangya Hospital, Central South University, Changsha 410008, China; 2Hunan Engineering Research Center of Skin Health and Disease, Xiangya Hospital, Central South University, Changsha 410008, China; 3Hunan Key Laboratory of Skin Cancer and Psoriasis, Xiangya Hospital, Central South University, Changsha 410008, China; 4National Clinical Research Center of Geriatric Disorders, Xiangya Hospital, Central South University, Changsha 410008, China; 5Department of Dermatology, Hunan Aerospace Hospital, Changsha 410205, China; 6Department of Social Medicine and Health Management, Xiangya School of Public Health, Central South University, Changsha 410008, China

**Keywords:** melanoma, delay in seeking medical attention, diagnostic delay, patient delay, Chinese

## Abstract

Melanoma is a highly malignant skin tumor, and prolonged delay in seeking medical attention (DSMA) and delay in diagnosis (DD) may result in poor prognoses. Through a web-based questionnaire, we explored the related factors affecting the DSMA and DD of melanoma in a Chinese population. A total of 112 valid answer sheets were received. After obtaining the relevant information, we analyzed the factors associated with DSMA and DD. The median time of DSMA was 8.0 (quartiles: 1.0, 29.3) months, and the median of patients’ DD was 1.0 (quartiles: 1.0, 8.3) month. The subsequent analysis showed that DSMA and DD were positively correlated to age and negatively correlated to education background and annual household income. Patients with a history of tumors or previous health-seeking behavior because of other skin lesions had significantly longer DSMA than those without. Patients who sought medical help at general tertiary hospitals for the first time had a significantly shorter DD than those who chose other hospitals. Our study found that DSMA and DD are associated with factors such as age, education, income, and patients’ histories. Secondary prevention of Chinese melanoma should be strengthened to reduce DSMA and DD to improve patients’ prognoses.

## 1. Introduction

Melanoma is a highly malignant cancer [1]. The incidence of melanoma continues to increase every year, accounting for a large proportion of skin cancer-related deaths [2]. In China, there are about 20,000 new cases of melanoma each year, and the number continues to rise annually [3]. Melanomas in China and Western countries are widely divergent. In the Caucasian population, the primary type of melanoma is cutaneous melanoma related to cumulative solar damage [4]. Melanoma in individuals with skin of color (SOC) is associated with a later stage and worse prognosis compared to melanoma in non-Hispanic white individuals, and the subtypes of melanoma in individuals with SOC are mainly acral melanoma and mucosal melanoma [5]. According to research on 522 Chinese melanoma patients [6], over 90% of patients are at stage II or above at the time of diagnosis, which may be because of multiple factors, including the inconspicuous presentation of lesions, mild subjective symptoms, and a lack of basic understanding of the disease.

Melanoma has a poor prognosis. In western studies, the 5-year survival rate of different stages of melanoma decreased from 91.4% (stage I) to 24.6% (stage IV), and non-ulcerative and Breslow thickness <1 mm melanoma have the best prognosis [7,8]. In China, the melanoma patients’ 5-year survival rate and median survival time were 41.6% and 43 months, and the 5-year disease-free survival (DFS) rate and median DFS time were 12.3% and 20 months, respectively [9]. Compared with Western patients, Chinese melanoma patients have the characteristics of later diagnosis and poor prognosis [6]. As early diagnosis improves the prognosis for cancer patients [10], early detection and medical treatment are essential for melanoma diagnosis and treatment in China.

We hypothesize that a prolonged delay before clinical diagnosis contributes to a later stage and poorer prognosis of Chinese melanoma patients. We believe that the delay before clinical diagnosis includes delay in seeking medical attention (DSMA) and delay in diagnosis (DD). DSMA refers to the time in months when the patient first became concerned about the mark of interest and when a doctor examined it [11], and DD means the interval between the first time that an individual patient seeks medical attention and the time that a pathological diagnosis of melanoma is made [12].

In this study, we analyzed the related factors affecting DSMA and DD based on a cross-sectional questionnaire among outpatient melanoma patients in a hospital in Hunan, China, and discussed corresponding suggestions to shorten the time of DSMA and DD.

## 2. Materials and Methods

### 2.1. Research Design

We adopted a cross-sectional study design at the department of dermatology in Xiangya Hospital, Central South University in Hunan, China. A total of 116 patients diagnosed with malignant melanoma by pathology were selected from March 2019 to December 2020.

### 2.2. Investigation Contents

When patients diagnosed with melanoma visit the outpatient clinic of the hospital, a professional dermatologist communicates with patients. After reading the informed consent form, the patient will choose whether to voluntarily complete the internet-based questionnaires (Appendix A). The contents included: general information (sex, age, education background, annual household income, and marital status), clinical characteristics (lesion site, age of onset, lesion morphology, lesion size, symptoms, previous diagnosis of other malignant tumors, and previous health-seeking behavior because of other suspicious lesions), and related delays (DSMA, DD).

### 2.3. Statistical Method

The Kolmogorov–Smirnov test was performed on the data, and the result showed that data such as DSMA and DD were consistent with the skewness distribution, so the data was described by the median and quartile. Between-group differences in DSMA and DD for unordered categorical variables were analyzed using the Mann–Whitney U test and Kendal’s Tau analysis were used to analyze the univariate correlation between DSMA, DD, and other variables. Since the data for DSMA and DD were heavily skewed with long left-sided tails, we then set age and gender as covariates and performed generalized linear models with a log-link and gamma response distribution to estimate the associations of statistically significant variables in univariate analysis with DSMA and DD. Data analysis and graphing were performed using SPSS 26.0.

### 2.4. Ethical Statement

All patients were asked to provide informed consent before inclusion. The study was approved by the Ethical Committee of Xiangya Hospital, Central South University (approval number: 202002024).

## 3. Results

A total of 112 valid answer sheets were received (excluding four invalid ones with missing data). Among them were 61 males (54.5%) and 51 females (45.5%); the mean age was 56.8 ± 14.4 years. The education background, annual household income, clinical characteristics, and related delays of the patients were shown in Table 1. There were 83 (74.1%) acral melanoma patients. The median (25–75th percentile range) of DSMA of all patients was 8 (1.0–29.3) months. Among them, 67 (59.8%) patients had a DSMA for over three months (not shown in tables). The median (25–75th percentile range) of patients’ DD was 1 (1.0–8.3) month.

The results of the Kendal’s Tau analysis and Mann–Whitney U test were displayed in Table 2 and Table 3; scattergrams and box plots between the corresponding variables were shown in Appendix A. The results showed that DSMA was positively correlated to age (τ = 0.201, *p* = 0.002) and negatively correlated to education background (*τ* = −0.217, *p* = 0.003) and annual household income (*τ* = −0.173, *p* = 0.016). The median and quartiles of DSMA in those patients who had been previously diagnosed with other malignant tumors or who had previous health-seeking behavior because of other suspicious lesions being significantly longer than those who had not (*p* < 0.05). As for DD, Kendal’s Tau analysis showed a positive correlation between DD and age (*τ* = 0.173, *p* = 0.012) and a negative correlation between DD and education background (*τ* = −0.192, *p* = 0.011) and annual household income (*τ* = −0.303, *p* = < 0.001). The median and quartiles of DD in patients with previous tumor history (*p* = 0.005) and who first visited other hospitals (*p* = 0.036) were significantly longer than in the other group.

Before we performed the gamma regression with log-link, we combined some of the categories in the variables to improve the test performance due to the small sample size. The results (Appendix A) showed patients who had been previously diagnosed with other malignant tumors (*p* = 0.06) or had previous health-seeking behavior because of other suspicious lesions (*p* = 0.08) may have had a longer DSMA than those who had not. The time of DD may be shorter in patients with better household income (*p* < 0.01) or no previous history of tumor (*p* = 0.02).

## 4. Discussion

The present analyses are based on a cross-sectional study at the department of dermatology in a general hospital in Hunan, China. We explored the influencing factors of related delay in melanoma patients. We found that DSMA and DD were correlated with age, education level, income, former medical history, and type of hospital first visit.

In other skin cancers, such as squamous cell carcinoma of the skin, men have significantly longer delay than women [13]. This phenomenon may be because women often pay more attention to appearance and are more aware of their body changes, making them more likely to detect skin lesions earlier [14]. Through the analysis, we observed that the median and quartiles of DSMA for women were shorter than those for men, but no statistical significance was found between genders in DSMA or DD.

In this study, DSMA and DD were positively correlated with age, which may be due to the insensitivity of elderly patients to changes in skin and symptoms and poor health awareness. In terms of education, DSMA and DD were negatively correlated with educational background. This may be because people with higher education levels have higher medical and health literacy and a better understanding of melanoma [15]. Therefore, highly educated people pay more attention to the changes in skin lesions. This result is consistent with the finding in multiple studies [16,17] that high literacy patients are more likely to be diagnosed with cutaneous melanoma at an earlier stage and with a thinner Breslow thickness than patients with a lower education background. In the economic aspect, DSMA and DD were negatively correlated with the annual household income. This may be related to the fact that low-income families have more considerable financial burdens on medical treatment. The lack of medical conditions in the living area also leads to the inability of patients to seek timely medical treatment. Given that education and annual household income are both important indicators of socio-economic status, people with higher education are more likely to have higher incomes, and that the DSMA is negatively correlated with both, we conducted a correlation analysis of these two variables, and the results did show a positive relationship (τ = 0.423, *p* < 0.001, not shown in tables).

As for the subtypes of melanoma, the result showed that the median and quartiles of DSMA and DD in acral melanoma patients are longer than other subtypes. This may be because changes in acral melanomas are less noticeable to patients than melanomas that grow elsewhere. Although no statistical significance was found in the subsequent analysis, we believe more research is needed to further explore this phenomenon.

Surprisingly, those previously diagnosed with other malignant tumors and who had previous health-seeking behavior because of other suspicious lesions had a longer DSMA than those who had not. This phenomenon may be because these people did not pay attention to their skin changes or had wrong perceptions of skin lesions due to their past consultant history. Therefore, after finding the lesions, they did not go to the hospital in time to seek medical treatment. This suggests that patients with a previous history of tumors or suspicious rashes should pay more attention to secondary prevention.

Of the 112 melanoma patients in this study, 67 (59.8%) experienced a delay in presentation (DSMA longer than three months [7]), and the DSMA of these 67 patients ranged from 4 months to 361 months. The longer the DSMA, the longer the progression of the disease and the delay in treatment. In related studies on breast cancer [18,19,20] and oral and throat cancer [21], patients diagnosed with advanced-stage (T3-T4 stage) had significantly more patient delay than early-stage (T1-T2 stage) patients. There is no doubt that the prognosis of advanced-stage patients was worse. Montella et al. [22] found a significant positive correlation between delay and tumor thickness but more studies [23,24] had shown no significant correlation between the delay and the tumor thickness of cutaneous melanoma, which may be because Caucasian melanomas are mainly CSD-related melanoma, with slower progression of skin lesions and a long horizontal growth period. However, Chinese melanoma and Western melanoma are quite different in terms of invasiveness and growth pattern, so the relationship between DSMA and tumor thickness needs further exploration.

Melanoma in the Chinese population is mainly acral and mucosal melanoma [25], and the skin lesions are primarily located in areas with less light exposure. Current studies have shown that acral melanoma and mucosal melanoma seem to be independent of UV exposure [25,26], so primary prevention measures, such as sunscreen, may not be as feasible as secondary prevention. As an essential method of secondary prevention, reducing the DSMA and DD cannot reduce morbidity but can improve the five-year survival rate and lower the mortality rate. Compared with Western countries, the Chinese had less understanding of melanoma and received a relatively late public education. According to Wu et al. [27], although the Chinese public has a positive attitude towards melanoma prevention and popular science education, they still lack basic knowledge of this disease. In conclusion, public health literacy should be improved to reduce DSMA. Oliveria et al. [9] found that increasing public awareness of melanoma can reduce the delay of patients. Korta et al. [28] believe melanoma-related education should be strengthened regardless of race. Since the 1960s, Queensland, Australia, has begun to educate the public about skin melanoma and has achieved positive results [29]. In terms of propaganda and education, the distribution of leaflets and online/offline public courses can be used to popularize the public’s understanding of melanoma and to understand its characteristics and hazards of it.

For melanoma patients and high-risk groups, Friedman et al. [30] introduced skin self-examination (SSE) in detail, which helps this group of people better observe their skin lesions and changes. Robinson et al. [31] found that training melanoma patients/high-risk individuals and their partners to perform SSE facilitates early detection of the second primary/primary melanoma. Therefore, for this part of the population, it is not only necessary to popularize melanoma-related knowledge but also to teach patients how to perform SSE. If suspected skin lesions are found, they should go to the hospital as soon as possible to improve the prognosis.

Different first-visit hospitals also impact DD; the median and quartile of DD for patients who went to third-class hospitals for the first time were shorter than those who chose hospitals of different tiers. Since most primary hospitals in China do not have sufficient medical conditions for diagnosing and treating melanoma, sending patients to a superior hospital as soon as possible can shorten the DD so that treatment can be carried out as quickly as possible. At the same time, it is necessary to carry out special training for grassroots doctors. When encountering suspected melanoma skin lesions, informing the patients in a timely manner and referring them to higher-level hospitals, which can also reduce the misdiagnosis rate and thus reduce the DD.

In this study, we conducted a detailed questionnaire survey on Chinese melanoma patients, obtained and described their basic information, clinical subtypes, and associated delay times, and initially found the factors that may affect their DSMA and DD. Our study had limitations. First, since this study is single-center, the sample selection may not be comprehensive enough. Second, due to the limitations of the research format, patient prognosis information was not recorded. Third, a possible risk of recall bias in self-report questionnaires cannot be excluded. Therefore, more large-scale cohort studies are needed to verify the effect of DSMA and DD on melanoma.

## 5. Conclusions

Our study found older, less educated, and lower-income people with a history of other malignancies are more likely to experience a longer DSMA and DD. Patients who had previous health-seeking behavior because of other suspicious lesions had a longer DSMA, and those who sought medical help at better hospitals for the first time had a shorter DD. For people with the aforementioned characteristics, secondary prevention of melanoma should be strengthened, such as improving disease education, practicing skin self-examination, and organizing skin disease screenings to shorten the delay.

## Figures and Tables

**Table 1 ijerph-19-14916-t001:** Patient characteristics.

**Characteristic**	**Overall, *n* (%)**	**Age, Years**	**Delay in Seeking Medical** **Attention, Month**	**Delay in** **Diagnosis, Month**
Total	112 (100)	57.0 (48.0, 67.8)	8.0 (1.0, 29.3)	1.0 (1.0, 8.3)
Sex				
Male	61 (54.5)	57.0 (50.0, 68.0)	8.0 (1.5, 36.0)	1.0 (2.0, 7.5)
Female	51 (45.5)	58.0 (41.0, 67.0)	6.0 (1.0, 20.0)	1.0 (1.0, 9.0)
Education background				
Primary school and lower	30 (26.8)	66.5 (58.0, 71.5)	14.0 (2.8, 51.8)	2.0 (1.0, 21.5)
Junior middle school	30 (26.8)	56.5 (47.8, 67.3)	11.0 (3.0, 31.5)	2.0 (1.0, 14.0)
Senior middle school	26 (23.2)	57.0 (48.8, 65.0)	5.5 (1.0, 60.0)	2.0 (1.0, 7.5)
College and above	26 (23.2)	44.0 (35.3, 59.0)	3.5 (0, 12.0)	1.0 (0, 2.0)
Annual household income				
Under 10,000 CNY	18 (16.1)	68.0 (57.0, 75.5)	20.0 (2.5, 129.0)	4.5 (1.0, 138.0)
10,000–29,000 CNY	18 (16.1)	58.5 (50.0, 68.3)	14.0 (6.0, 25.5)	2.5 (1.0, 15.5)
30,000–49,000 CNY	31 (27.7)	58.0 (49.0, 66.0)	3.0 (1.0, 24.0)	2.0 (1.0, 9.0)
50,000–99,000 CNY	22 (19.6)	55.0 (48.0, 65.5)	2.5 (0.8, 19.8)	1.0 (1.0, 2.3)
Above 100,000 CNY	23 (20.5)	50.0 (36.0, 58.0)	6.0 (1.0, 18.0)	1.0 (0, 2.0)
Marriage				
Unmarried or divorced	15 (13.4)	64.0 (39.0, 66.0)	3.0 (1.0, 12.0)	1.0 (1.0, 2.0)
Married	97 (86.6)	57.0 (48.0, 68.0)	9.0 (1.0, 36.0)	2.0 (1.0, 12.0)
Melanoma subtype				
Others	29 (25.9)	50.0 (39.0, 58.5)	3.0 (0.5, 30.0)	1.0 (1.0, 2.0)
Acral melanoma	83 (74.1)	59.0 (52.0, 68.0)	10.0 (2.0, 30.0)	2.0 (1.0, 13.0)
Previous diagnosis of other malignant tumors				
Negative	86 (76.8)	57.0 (47.8, 66.0)	6.0 (1.0, 24.0)	1.0 (1.0, 4.5)
Positive	26 (23.2)	59.0 (52.8, 69.8)	17.5 (2.8, 75.0)	1.0 (4.5, 22.5)
Previous health-seeking behavior because of other suspicious lesions				
Negative	87 (77.7)	47.0 (57.0, 67.0)	4.0 (1.0, 24.0)	1.0 (1.0, 6.0)
Positive	25 (22.3)	59.0 (48.0, 68.0)	18.0 (6.0, 54.5)	3.0 (1.0, 30.5)
Type of hospital first visit				
General tertiary hospitals	58 (51.8)	56.0 (47.0, 66.0)	6.0 (1.0, 18.0)	1.0 (1.0, 3.0)
Other hospitals	54 (48.2)	58.0 (49.0, 69.0)	11.0 (1.8, 39.8)	2.5 (1.0, 17.0)

Data is median (25–75th percentile range) unless otherwise specified.

**Table 2 ijerph-19-14916-t002:** Kendal’s Tau analysis between DSMA and DD and variables.

Variables	DSMA		DD	
	*τ*	*p*	*τ*	*p*
Age	0.201	**0.002**	0.173	**0.012**
Education background	−0.217	**0.003**	−0.192	**0.011**
Annual household income	−0.173	**0.016**	−0.303	**<0.001**

Bold text denotes significance (*p* < 0.05). DSMA, delay in seeking medical attention; DD, delay in diagnosis. *τ*: Kendal’s Tau coefficient.

**Table 3 ijerph-19-14916-t003:** Mann–Whitney U test between DSMA and DD and variables.

Variables	DSMA	DD
	*p*	*p*
Sex	0.273	0.274
Marriage	0.116	**0.045**
Melanoma subtype	0.129	0.080
Previous diagnosis of other malignant tumors	**0.015**	**0.005**
Previous health-seeking behavior because of other suspicious lesions	**0.006**	0.257
Type of hospital first visit	0.272	**0.036**

Bold text denotes significance (*p* < 0.05). DSMA, delay in seeking medical attention; DD, delay in diagnosis.

## Data Availability

Not applicable.

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
