# Peer review of "Delay in Seeking Medical Attention and Diagnosis in Chinese Melanoma Patients: A Cross-Sectional Study"

_ijerph, 2022, doi:10.3390/ijerph192214916_

Round 1

Reviewer 1 Report (Previous Reviewer 2)

Dear Authors

You have improved the quality of your study, mainly the statistical analysis.

Congratulations. 

Kind regards

Reviewer 2 Report (Previous Reviewer 3)

Authors have satisfactory addresses the comments.

This manuscript is a resubmission of an earlier submission. The following is a list of the peer review reports and author responses from that submission.

Round 1

Reviewer 1 Report

Thank you for the opportunity to review this interesting manuscript. 

The authors have conducted a study related to melanoma in a Chinese population.  Of 116 questionnaires issued to melanoma patients in a Chinese treatment centre, 112 valid responses were returned. A mean delay in seeking medical attention (DSMA) was 29.3 months (Median 8) is reported.  A median delay in diagnosis (DD) of 1 was also reported.   Positive correlations were with age, and negative correlations were with education and income.  A history of tumour or other lesions was associated with longer DSMA.  Seeking help at a tertiary hospital was associated with shorter DD than at other hospitals.  

The study is interesting and should be published.  However, I do think that there are a number of aspects that do require significant improvement and re-analysis before the work is suitable for the press.  I very much hope that the authors make the requested changes below, so that the work can be made appropriate for reporting.  

In general, the paper is well written with good English, although there are some places where English does need to be corrected and the places where I noticed this are briefly recorded below.   There are a number of separate minor separate points, that the authors may wish to consider and address.  

Of more concern, are that: some of the data has been analysed with statistical tests that are not appropriate for the type of data collected (Spearman's Rho instead of Kendal's Tau; Wilcoxon's Ranked Sign Test instead of Mann Whitney U Test); and some results for median and 25th percentile values are equivalent, which seems numerically impossible. 

Also, scattergrams showing the actual values are missing, and should appear in at least Supplemental Materials.  

Please find detailed comments on these aspects and others I do think are important below. 

If the authors are able to address all of the concerns raised below, I do think that the work would be suitable for publication. 

TITLE

Would be more clear if rephrased to:    Delay in seeking medical attention and diagnosis in Chinese melanoma patients: A cross sectional study

ABSTRACT

Line 2 - 'of Chinese melanoma' sounds like a discrete pathological condition.  Instead, the term used should be 'of melanoma in China'.  Given that this is data from a single centre, it might be more correct to say 'of melanoma in a Chinese population'. 

The term 'other rashes' seems inappropriate, because melanoma is not a rash.  A possibly more correct term might be 'other skin lesions'. 

INTRODUCTION

Line 44: The phrase - 'which may be caused by multiple causes such as inconspicuous rashes, mild subjective symptoms, and a lack of basic understanding of the disease'. is slightly confusing, because it suggests these are the causes of the melanoma.  A wording that would be more clear would be 'which may be because or multiple factors, including the inconspicuous presentation of lesions, mild subjective symptoms, and a lack of basic understanding of the disease'. 

Line 60: The sentence containing the phrase 'the first time seeking medical attention ...', would read better as: 'the first time that an individual patient seeks medical attention, and the time that a pathological diagnosis of melanoma is made'.  

MATERIALS AND METHODS

Line 75:  'rashes' should be replaced with the term 'lesions'. 

There is no clear description of how patients were approached in the methods section.  Were patients approached by clinicians in the clinics, or was this by phone or mail?

The survey used should be provided in supplemental materials in both the original Mandarin, and in an English translation, so that the reader can see how the data was collected.  

Spearman's correlation was used to assess correlations, but this is a parametric test and the skewedness of data suggests a non-parametric distribution.  For this reason, Kendal's Tau would seem a more appropriate measure for correlation in the current data. 

The Wilkoxon's Ranked Sign Test was used to compare paired correlates.  Unfortunately, this test is specifically for paired data where each individual acts as their own control.  Unless I am mistaken, the data analysed is of unpaired data, where individuals who for example may be married, are compared with those who are not.  On that basis, the correct test to use would be the Mann Whitney U Test.

RESULTS 
Line 87: The past tense should be used 'There were 83' as opposed to the current 'There are'. 

In the abstract as well as in Table 1, some data for delay in diagnosis are given where the median is equivalent to the 25th percentile value (eg Total 1.0 (1.0, 8.3).  This seems to be a systematic error of some sort, because the median by definition cannot be equivalent to the 25th percentile value.  The only  exception that occurs to me, would be if all values are the same, but that is not the case because the 75th percentile value is given as substantially higher. 

While the correlation values shown in Table 2 are informative, and as mentioned above Kendal's Tau would be more appropriate, it would be further helpful to see the actual data distribution in scatter plots.  Those the authors wish to especially emphasise should be shown in the paper, while the remaining scatter plots should appear in supplemental materials. 

Data in Table 2 showing results for Correlation and opposing groups, should 1)  firstly apply tests appropriate to the data type (ei - Kendal's Tau instead of Spearman's correlation on basis that data are not normally distributed; and the Mann Whitney U Test instead of the Wilkoxon's Ranked Sign Test on basis that the groups compared are not truly paired, but are of different subjects);  and 2) be divided into two tables, with Table 2 showing just correlation data, and Table 3 showing the opposing group analysis data, the reason being that it is confusing and unhelpful to have spaces in the column for correlation values where none can exist in the opposing group results.  

DISCUSSION

Line 112: The word 'formerly' should be replaced with 'former', and 'etc' should be deleted and replaced with terms describing the specific variables of interest. 

Education and income are correctly identified as separate variables, however the discussion as written does not seem to recognize that these two variables are nonetheless related.  This should be briefly mentioned in the discussion, and also the manner in which similarity in results for the two variables might reflect their relation, should also be considered. 

Reviewer 2 Report

Attached.

Reviewer 3 Report

Though study is well designed, a detailed questionnaire is required to reach the solid conclusion. There are multiple factor that can influence the outcome such as the geological location of residents, distance/availability of hospital, family history of cancer etc.

Reviewer 4 Report

The authors presented an interesting report on melanoma diagnostic delay. However, some changes are needed to improve the quality of the article. 

- the abstract lacks background information

- it is not clearly stated where the study was performed (name of the hospital, division/section)

- some repetitions are present (e.g. "tertiary hospital") and should be avoided

- a statement regarding Ethical Committee approval and patient consent is needed

- English language revision is highly recommended